# Changes in Electrical Capacitance of Cell Membrane Reflect Drug Partitioning-Induced Alterations in Lipid Bilayer

**DOI:** 10.3390/mi14020316

**Published:** 2023-01-26

**Authors:** Shide Bakhtiari, Mohammad K. D. Manshadi, Mehmet Candas, Ali Beskok

**Affiliations:** 1Mechanical Engineering Department, Southern Methodist University, Dallas, TX 75275, USA; 2Department of Biological Sciences, University of Texas at Dallas, Dallas, TX 75080, USA

**Keywords:** microfluidics, dielectric spectroscopy, cell membrane capacitance, drug uptake, ion channel blockers

## Abstract

The plasma membrane is a lipid bilayer that establishes the outer boundary of a living cell. The composition of the lipid bilayer influences the membrane’s biophysical properties, including fluidity, thickness, permeability, phase behavior, charge, elasticity, and formation of flat sheet or curved structures. Changes in the biophysical properties of the membrane can be occasioned when new entities, such as drug molecules, are partitioned in the bilayer. Therefore, assessing drugs for their effect on the biophysical properties of the lipid bilayer of a cell membrane is critical to understanding specific and non-specific drug action. Previously, we reported a non-invasive technique for real-time characterization of cellular dielectric properties, such as membrane capacitance and cytoplasmic conductivity. In this study, we discuss the potential application of the technique in assessing the biophysical properties of the cell membrane in response to interaction with amiodarone compared to aspirin/acetylsalicylic acid and glucose. Amiodarone is a potent drug used to treat cardiac arrhythmia, but it also exerts various non-specific effects. Compared to aspirin and glucose, we measured a rapid and higher magnitude increase in membrane capacitance on cells under amiodarone treatment. Increased membrane capacitance induced by aspirin and glucose quickly returned to baseline in 15 s, while amiodarone-induced increased capacitance sustained and decreased slowly, approaching baseline or another asymptotic limit in ~2.5 h. Because amiodarone has a strong lipid partitioning property, we reason that drug partitioning alters the lipid bilayer context and subsequently reduces bilayer thickness, leading to an increase in the electrical capacitance of the cell membrane. The presented microfluidic system promises a new approach to assess drug–membrane interactions and delineate specific and non-specific actions of the drug on cells.

## 1. Introduction

The plasma membrane of cells is a lipid bilayer containing various types of proteins embedded to form a 3-dimensional mosaic assembly [1]. The lipid bilayer comprises amphipathic phospholipids that face the cell’s interior and exterior with their water-soluble polar head groups [2]. At the same time, their water-insoluble hydrophobic fatty acid chains are tucked into the hydrophobic core portion. While the bilayer establishes the boundary of the cell, it is also structurally and functionally an integral part. It serves as an interface where the cell interior communicates with the external environment. Thus, cell survival depends not only on the ability of the plasma membrane to protect the cell interior and membrane-associated functions involved in sensing environmental changes, but also recognizing and transducing extracellular molecular signals, controlling the influx and efflux of ions and molecules, and mediating cellular responses [3,4].

Changes in the biophysical properties of the membrane affect molecular interactions and protein function within the membrane by altering lateral and transverse forces within the lipid bilayer [5]; such changes can be occasioned by varying the lipid composition in the bilayer leaflets and when new entities, such as drug molecules, are partitioned in the bilayer [6]. In the case of drug partitioning, the bilayer can undergo dimensional adjustments in the presence of partitioned molecules due to its fluidic nature [7]. In fact, before interaction with their desired cellular targets, many drug compounds go through an initial partitioning in the lipid bilayer membrane. Partitioning in the lipid bilayer can cause a sink effect, markedly influencing the binding of the drug molecule to its target and compromising its potency [8]. Significantly, drug partitioning in the lipid bilayer can affect the biophysical properties of the bilayer, leading to intramembrane dimensional changes occasioned by intra-bilayer physical accommodations and altered interactions between bilayer components [9]. Even drug compounds targeting intracellular targets can induce dimensional alterations in the membrane if they partition in the lipid bilayer [10]. Such dimensional accommodations in the bilayer structure can complicate the desired pharmacological action of drugs, as they would accompany non-specific alterations in the structure and function of many membrane proteins and associated cellular events. Therefore, assessing drugs for their effect on the biophysical properties of the lipid bilayer of a cell membrane is critical to understanding specific and non-specific drug action. The present study employed the dielectric spectroscopy technique to investigate drug effects on the lipid bilayer.

Dielectric spectroscopy (DS) is a non-invasive, promising, and fast method for finding dielectric properties of any material, such as biological cells, under AC voltage and desired frequency domain in real time. Previously, our group developed a microchip consisting of microwell arrays on the electrode surface as a first step [11]. Positive dielectrophoresis and negative dielectrophoresis were applied to yeast cells to capture and release the cells from microwells. It was the first microchip that combined DS with electrodes with microwells to study biological cells [11]. In a further study, the microfluidic chip was tested for a human prostate cancer cell line, PC-3, suspended in low conductivity buffers (LCB). Dielectric properties of the cells, such as cytoplasmic conductivity and membrane capacitance, were extracted from impedance spectra using an equivalent circuit model [12]. The viability of PC-3 cells under 100 μM Enzalutamide drug treatment was also tested [13]. However, the major drawback of the chip was the Electrode polarization (EP) effect. EP is the accumulation of ions and charged particles at the electrode/electrolyte interface. Deposition of gold nanostructures (GNs) on the surface of electrodes can minimize EP, which is dominant at low frequencies and high conductive media [14,15]. In our most recent study, DS was applied on PC-3 cell suspension in growth media (high conductive media), which was captured in microwells with dendritic nanostructured gold surfaces. Dielectric properties of the cells were extracted to show the stability of the cells in the microfluidic system by using physiological media [16].

It was demonstrated that voltage-gated ion channels on the PC-3 cells play a significant role in cell growth. One of the methods to decrease PC-3 cell proliferation is inhibiting these channels’ function [17]. Ion channel blockers and openers are used as a treatment, increasing growth and inhibition. Dequalinium, amiodarone, minoxidil, and diazoxide are some effective drugs used for this purpose [18]. Amiodarone is a multichannel blocker (K^+^, Ca^+^, and Na^+^) and the most effective one for blocking ion channels and growth inhibition. The effectiveness of amiodarone for 1 µg/mL to 10 µg/mL concentrations on human cancer prostate cells was studied. The results prove that an amiodarone concentration of more than 1 µg/mL significantly inhibited prostate cancer cell growth, while the PC3 cell viability was not affected within the first hour of the experiments [19].

In our recent study, we reported a microfluidic impedance spectroscopy technique that enables the measurement of dielectric properties of biological cells in high-conductivity physiological buffers [16]. Specifically, this non-invasive technique captures individual cells on an electro-activated microwell array, which enables real-time characterization of cellular dielectric properties, such as membrane capacitance and cytoplasmic resistance. In this study, we discuss the potential application of the technique to measure the responses of PC-3 prostate cancer cells to different doses of amiodarone, aspirin, and glucose. Cell responses to these drugs were recorded as a function of time, and cell membrane dielectric properties, including the membrane capacitance and cytoplasmic resistance, were extracted from the equivalent circuit model.

## 2. Material and Method

### 2.1. Experimental Setup

The experimental setup is shown in Figure 1A. The suspended PC-3 cells in RPMI-1640 were injected from the inlet to the microchannel and captured with gravitational force [16]. The syringe pump injected the growth media with a flow speed of 3 µL/min to wash uncaptured cells and nourish the cells inside the wells during the measurements. The microfluidic chip was connected to the impedance analyzer (HP Agilent 4194A, Keysight Technologies, Santa Rosa, CA, USA) to record the device impedance from the 10 kHz to 40 MHz frequency range at 20 mV AC voltage. The impedance data were taken every ten seconds, and after reaching the steady state condition, the drug was applied to the device using a syringe pump. RPMI-1640 was also used to dilute amiodarone with different concentrations and Dulbecco’s Modified Eagle’s Medium (DMEM) (Sigma-Aldrich, St. Louis, MO, USA) for making 10 µM aspirin and 5.5 mM glucose solution. Injecting the drug was repeated for amiodarone concentrations of 10 µg/mL, 5 µg/mL, 4 µg/mL, 3 µg/mL, 2 µg/mL, and 1 µg/mL. The same procedure was repeated for 10 µM aspirin and 5.5 mM glucose.

The impedance data from the measurements were transferred to a PC and analyzed using an in-house MATLAB algorithm to extract parameters from the equivalent circuit model shown in Figure 1B, which represents the entire microchip as a collection of multiple parallel sub-elements consisting of filled and empty microwells, and a parallel device stray capacitance *C_f_*. Each sub-element has the same channel resistance *R_ch_*. The electrode polarization is modeled using the constant phase element (*CPE*) model with an impedance of $Z_{CPE}=1/\left[K\;\left(j\omega^{\alpha }\right)\right]$, where *α* and *K* are the exponent and the coefficient, *ω* is the frequency, and $j=\sqrt{-1}$. The empty and filled wells have resistances of *R_W,e_* and *R_W,f_*, respectively, and the cells in filled wells are represented by their cytoplasm resistance *R_cyt_* and cell membrane capacitance *C_mem_*. The cell membrane is a shield to the cytoplasm and acts as a separator between the two conductors (media and cytoplasm). Therefore, the membrane is considered a capacitor [20]. Additionally, the cell membrane resistance, if modeled as a parallel RC circuit, is on the order of 1 MΩ [21]. Therefore, perceivable resistance changes in the cell membrane do not significantly affect the impedance data in the considered range of frequency (10 kHz–40 MHz). The *SU*8-covered well structure is modeled as a capacitive layer with $C_{SU8}=\varepsilon_{r}\varepsilon_{0}A/t$, where $\varepsilon_{r}$ is the relative permittivity of *SU*8, and *A* and t are the surface area and thickness of the wells, respectively. Extraction of the circuit model parameters ($R_{ch}$, *R_W,e_*, *R_W,f_*, K, α, and $C_{f}$, $R_{cyt}$, and $C_{mem}$) from the complex impedance data are achieved by using nonlinear least square method data fit with the experimental data using the Marquardt–Levenberg optimization algorithm. In order to have consistency, 10 random sets of initial conditions were used in the algorithm. The average values are reported as the results when the data from the initial conditions converge to a single global solution set. The quality of the fit was measured by the coefficient of determination (R^2^) with values above 0.99 [22]. The fitting procedure was applied in a sequential manner to growth media (GM), cells in growth media (CGM), growth media with drug (GMD), and cells in growth media with drug (CGMD) as explained in [10]. The methodology used for finding the equivalent circuit variables can be summarized as follows:

For growth media (GM) parameters, fitting to the impedance data were used to find $R_{ch}$, $R_{w,e}$, K, α, and $C_{f}$.

For cells in growth media (CGM), the impedance data are used to find all 8 parameters. However, the values for $R_{ch}$ and *R_W,e_* are bounded with ±5% variation from their Step 1 values, and the values for K, α, $C_{f}$, *R_W,f_*, $R_{cyt}$, and $C_{mem}$ are found.For GM with drug (GMD), the electrical circuit parameters ($R_{ch}$, *R_W,e_*, K, α, and $C_{f})$ are found using the procedure in Step 1.For cells in growth media with drug (CGMD), the impedance data are used to find all 8 parameters. However, the values for $R_{ch}$ and *R_W,e_* are bounded with ±5% variation from their Step 3 values, and the values for K, α, $C_{f}$, *R_W,f_*, $R_{cyt}$, and $C_{mem}$ are found.To find the cell parameters for all subsequent time steps in CGM or CGMD stages, $R_{ch}$, *R_W,e_*, K, α, and $C_{f}$ are bounded with ±5% variation from their Step 5 values and *R_W,f_*, $R_{cyt}$, and $C_{mem}$ are obtained.

### 2.2. Device Fabrication

The proposed device in this paper consists of two 1 mm × 1 mm square electrodes that align on top of each other and 30 μm × 30 μm × 30 μm microwells in between. To fabricate this device, gold electrodes were prepared by the photolithography method. First, glass slides were cut to 2.5 cm × 2.5 cm and then washed to clean the electrode substrate. An ultrasonic bath (FB11201, Fisher Scientific, Pittsburgh, PA, USA) was used to wash glasses in four steps with 1 M KOH, acetone, isopropyl alcohol, and deionized (DI) water, respectively, for 10 min each step. Right after, they fully dried at 150 °C in the oven. Then, in two steps, a positive photoresist (S1813) was coated on the cleaned and dried glass substrates using a spin coater (VTC-100PA, MTI Corporation, Richmond, CA, USA). In the first step, the coating speed was 1000 rpm for 10 s, and in the second step, the 4000 rpm was used for 3 s with 300 rpm/s acceleration/deceleration stages; the coated glasses were soft baked for 1 min at 115 °C on a hot plate. Afterward, they were exposed to UV light (110 mJ/cm^2^) for 11 s by a mask aligner (Karl Suss, MJB3, SUSS MicroTec, Garching, Germany) and a proper transparency mask. The substrates were submerged into an MF-26A developer for 17 s; the UV-treated regions were washed away. Finally, the substrates were placed in a chromium–gold sputter coater (EMS300TD, Emitech, Montigny-le-Bretonneux, France) to coat a layer of chromium and gold in 60 mA-60 s conditions. After sputtering chromium and gold, the substrates were developed into PG remover to lift off the extra gold-coated photoresist regions. The next step of device fabrication is producing microwells on the surface of the bottom electrode. To do that, a positive photoresist (SU8) was used on the square shape gold pat of the electrode with 1 mm × 1 mm dimensions. An array of 21 × 21 microwells with 30 μm × 30 μm × 30 μm dimensions was built.

The dendritic nanostructures were built on the bottom of the microwells using the electrochemical deposition method. A three-electrode potentiostat/galvanostat system (EZstatPro 7, Nuvant) at −0.7 volt was used for this purpose. Planar gold electrodes were immersed in 0.5 mg/mL sodium tetrachlorocuprate (III) (AuCl_4_Na_2_H_2_O) solution (Sigma-Aldrich, St. Louis, MO, USA), and the system was turned on for 60 min (Figure 1C). After this procedure, 70 μm-thick double-sided tape was used to fabricate the microchannel. The tape was cut with a craft cutter (Silver Bullet) to produce 1 mm × 15 mm channels. The inlet and outlet for the channel were created by drilling two holes on the top electrode sides using a diamond drill bit. Then, the top and bottom electrodes were aligned with a mask aligner. The bottom side had microwells with nanostructures, and the top had inlet and outlet holes. It should also be mentioned that each test was performed in triplicates, which required the fabrication of a new device.

### 2.3. Cell Preparation

The PC-3 cancer cells were obtained from the Urology Lab at UT Southwestern, Dallas, Texas. The cells were maintained in RPMI 1640 growth medium (Sigma-Aldrich, St. Louis, MO, USA) supplemented with 5% fetal bovine serum (FBS), penicillin (100 IU/mL), and streptomycin (100 μg/mL). The cells were grown in an incubator (Thermo Scientific, Pittsburgh, PA, USA) at 37 °C with 5% CO_2_. The culture was grown for 72 h to reach confluency (~90% coverage of the flask surface). Twenty-four hours before the experiments, the cells were treated with trypsin (TrpLE) for 5 min. Trypsin action was stopped by adding 6 mL complete medium (containing 5% FBS), and cells were collected by centrifugation (Thermo Scientific ST8, Pittsburgh, PA, USA) at 2000 rpm for 4 min at room temperature. The pellet was suspended in DMEM containing 5.5 mM glucose (Sigma-Aldrich, St. Louis, MO, USA) to reach 105 cells/mL cell density. For measurements with aspirin/acetylsalicylic acid treatment, the experiment was performed with 10 μM (0.01 mM) aspirin (Sigma-Aldrich, St. Louis, MO, USA) in a culture medium. This aspirin concentration is in the therapeutic dose range used in clinical settings. No adverse effects were reported for aspirin on cultured cell lines at 1–10 μM concentrations. Thus, aspirin at μM concentration is not expected to cause cell death, change the cell cycle, or induce apoptosis or necrosis. For experiments with glucose treatment, PC-3 cells were maintained in DMEM without glucose for 24 h until the measurements.

## 3. Results

In our previous study [16], we demonstrated that dendritic gold nanostructures provide the context to measure cell properties in a physiological buffer due to the significant reduction in the EP effect. Herein, we explored the potential use of the microfluidic device for real-time monitoring of cell membrane capacitance in response to different doses of amiodarone compared to aspirin/acetylsalicylic acid and glucose. The three compounds were selected for the study because they have distinct chemical, physical, and biological properties. Amiodarone is an antiarrhythmic medication used to treat and prevent certain types of life-threatening ventricular arrhythmias (abnormal heart rhythm). It is known to have a complex electrophysiological effect. Aspirin/acetylsalicylic acid is the most common non-steroidal anti-inflammatory drug. Glucose is the main six-carbon simple sugar in the blood and is utilized as a metabolic fuel in all cells.

### 3.1. Extracted Parameters for Drug Concentrations

Amiodarone was diluted in growth medium DMEM at different concentrations; 10 µg/mL, 5 µg/mL, 4 µg/mL, 3 µg/mL, 2 µg/mL, and 1 µg/mL, and aspirin (10 µM) and glucose (5.5 mM) were used as control measurements. Media containing amiodarone, aspirin, or glucose were applied into an empty device with no cell to extract *R_ch_*, *R_w,e_*, *K*, *α*, and *C_f_* (Table 1). These parameters constitute the device parameters for empty wells, which were kept constant for extracting the cell and filled well parameters afterward.

According to Table 1, the solution resistance *R_ch_* decreases slightly with increased amiodarone concentration. This result is consistent with the solution’s measured conductivity with increased amiodarone concentration. *R_ch_* has a small change for different doses of amiodarone, glucose, and aspirin. Due to the use of small amounts of drugs, *R_ch_* value was dominated by the growth media. Similarly, *R_w,e_*, which represents the resistance of the empty well, was constant for the drugs. Additionally, the parasitic effect (*C_f_*) remained nearly constant, as shown in Table 1. This is due to the consistency of the test experiment conditions. The table also shows K and α, which are the CPE coefficient and exponent, respectively. The K and α variations show opposite trends for amiodarone concentrations from 1 µg/mL to 10 µg/mL. In comparison, *K* increases by increasing the drug concentration from 1 µg/mL to 10 µg/mL, while α decreases. Considering that the value of α varies between 0 and 1, where 1 represents purely capacitive, and 0 corresponds to purely resistive behavior, reduction in *α* shows increased resistive response with increased doses of amiodarone, although these changes are quite small.

### 3.2. Extracted Parameters of Cells before and after Drug Injection

After extracting the electric properties of drug solutions, their effects on the PC-3 cancer cell were studied. Before applying the drug, growth media flowed through the device; then, the drug with the desired concentration was injected into the device. All system parameters were extracted before and after drug injection to study drug cell response. Table 2 shows cells and electric device properties before drug injection and 70 s after injection of different amiodarone doses, glucose, and aspirin. It is important to note that cell response after drug injection was transient. Therefore, the results presented in this table constitute the values obtained 70 s after drug injection. The membrane capacitance for prostate cancer cells were reported in a range from 18 (pF) to 150 (pF) [23,24,25,26,27]. Table 2 shows that the *C_mem_* is about 100 (pF) before drug injection, which is in the range suggested in the literature.

According to Table 2, *α* values decrease by increasing the amiodarone concentration, and the lowest value of *α* corresponds to 10 µg/mL concentration, while its maximum value corresponds to the cells before drug injection. *K* values show the opposite behavior of *α* values and increase by increasing the drug concentration. The changes for *K* and *α* for CGM and CGMD are similar to the trends of these parameters for different drug concentrations without cells (GMD). Since the growth media is the base solution for drug dilution, the resistivity of all studied buffers was almost constant. Therefore, the ranges of *R_ch_* and *R_W,e_* for all cell data fittings before applying the drug are selected to be in the range of that in GM. However, after applying the drug, these parameters are set to be in the range of the drug solution (GMD) shown in Table 1. After drug injection, the resistance of the filled well (*R_W,f_*), membrane capacitance (*C_mem_*), and cytoplasm resistance (*R_cy_*) were found at different drug concentrations (CGMD). A sample of these parameters 70 s after introducing the drug (400 s after starting the experiments) is summarized in Table 2**.** The results show no significant changes in the cytoplasm resistance with increased drug concentration. This is because amiodarone is a multi-ion channel blocker, which maintains the ionic conductivity of the cell’s constant. However, the membrane capacitance shows considerable variations with the drug dose 70 s after drug injection.

### 3.3. Effect of Drug on Membrane Capacitance and Resistance

The effects of amiodarone on the membrane capacitance (*C_mem_*) and cytoplasmic resistance (*R_cy_*) of PC-3 cells were measured as a function of time. Real-time measurements with varying concentrations of amiodarone, glucose (5.5 mM), and aspirin (10 µM) were collected for 1000 s. Before administration of the medium containing the compounds, growth medium/DMEM without any addition was flowed into the device for 330 s to reach a steady state in the measurements. This procedure was performed in all experiments to validate the effects of compounds on the cells. After establishing a steady state measurement, a medium containing the compound at the indicated concentrations was injected into the device. The compound effect on the captured cells inside the wells was recorded.

All three compounds (amiodarone, aspirin, and glucose) increased membrane capacitance immediately after introduction (Figure 2A). The increase in membrane capacitance was noticeably smaller for aspirin and glucose than the increase in capacitance occasioned by amiodarone, whose response was dose-dependent. The change in the cell membrane capacitance was maximum for 10 µg/mL amiodarone concentration, and this response was reduced by concentration reduction. While aspirin and glucose slightly increased the membrane capacitance, these values quickly returned to baseline (initial/pre-treatment level) in 15 s. Amiodarone-induced increased membrane capacitance exhibited a markedly slight decrease (asymptotes may reach baseline or to another limit at around 10,000 s; ~2.5 h). Importantly, the increase in capacitance occurred for all experiments without any significant changes in the cytoplasm resistance (Figure 2B), indicating no substantial movement of ions in or out of the cells. We reason that no change in the cytoplasmic resistance may be attributed to amiodarone inhibiting both inward and outward currents. Additionally, amiodarone partitioning in cell membrane and resultant membrane structural modulation that changes membrane capacitance may be a rectifying event, sustaining cytoplasmic resistance unchanged. Thus, there was no apparent change in overall cytosolic resistance due to interaction with the drug, and there was no apparent change in the cell size/volume, which could occur due to cell swelling.

The measurements for amiodarone-induced membrane capacitance change were repeated three times, and standard deviations extracted from the data were analyzed. Figure 3 shows cell membrane capacitance variation as a function of time for 10 µg/mL amiodarone. Although the data were obtained every 10 s, standard deviations from the measurements are shown every 50 s in order not to clutter the plot. The results validate the precision of measurements and data extraction procedures, as well as the repeatability of the experiments.

## 4. Discussion

We conducted amiodarone, aspirin, and glucose experiments to study their effects on membrane capacitance. Compared to aspirin and glucose, we measured a rapid and higher magnitude increase in membrane capacitance on cells under amiodarone treatment. Increased membrane capacitance induced by aspirin and glucose quickly returned to baseline in 15 s, while amiodarone-induced increased capacitance sustained and decreased slowly, approaching the baseline or another asymptotic limit in ~2.5 h.

The primary simplification in our electric circuit model is that it represents all changes that happen in and on the cell membrane, as well as the effects of the electric double layers (EDL) outside and cytosol regions using a single parameter, *C_mem_*. Data presented in Table 1 show nearly constant channel and microwell resistances for different amiodarone doses. This indicates that the ionic conductivity of the solution, and hence, the EDL thickness and its capacitance outside the cell membrane are nearly constants. Additionally, constant cytoplasm resistance indicates constant cytoplasm conductivity. Therefore, we do not expect large variations in the cytosol EDL capacitance. Overall, the equivalent circuit model for the cell is similar to the commonly used single shell model, and the measured complex impedance data fits extremely well to this circuit model without adding any other elements.

The reason for large changes in the measured membrane capacitance could be due to the aggregation of amiodarone molecules on the cell surface, followed by lipid partitioning. Capacitance is directly proportional to the cell surface area, A, and inversely proportional to the cell membrane thickness, d. Amiodarone has low water solubility with a topological surface area of 42.7 Å^2^. Attachment of the drug particles on to the cell surface (membrane proteins, polysaccharide, or lipid) can increase the net surface area, while lipid partitioning can reduce the membrane thickness. Both effects will increase the measured cell membrane capacitance in the equivalent circuit model, which does not consider the contributions of the drug separately. It is important to emphasize that amiodarone has high lipid partitioning (LogP = 7.9), which leads to dynamic interaction and partitioning in the lipid bilayer. Because amiodarone has a strong lipid partitioning property, we reason that drug partitioning alters the lipid bilayer context and subsequently reduces bilayer thickness, leading to an increase in the electrical capacitance of the cell membrane. We posit that membrane capacitance is a useful parameter for assessing how drug partitioning in the lipid bilayer can change the biophysical properties of the lipid bilayer because dimensional changes occasioned by drug interactions and partitioning in the cell membrane can affect the function of membrane proteins as well as cellular responses associated with their functionality.

Amiodarone is recognized as an effective drug for the treatment of heart arrhythmia. Furthermore, along with the inhibition of potassium channels, amiodarone is also known to interfere with the function of several other membrane proteins, including calcium and sodium ion channels, as well as adrenergic receptors in cells responding to epinephrine, norepinephrine, and dopamine. Thus, ionic mechanisms involving the action of amiodarone in antiarrhythmic treatment and its simultaneous promiscuous action on various membrane proteins are not well understood. We measured an increase in cell capacitance when cells interacted with amiodarone. The increase in capacitance occurred without any change in resistance, indicating no apparent change in cell size/volume, which could occur due to cell swelling. Thus, there was no apparent change in overall cytosolic resistance due to interaction with the drug (Figure 2B). We combine our interpretation in a graphic model that an increase in cell membrane capacitance is occasioned by drug partitioning in the lipid bilayer, which results in dimensional changes in bilayer thickness and alters the structure and function of proteins associated with the membrane (Figure 4A). In our circuit model, the bilayer structure of the cell membrane has capacitance (*C_mem_*), which is serially linked to cytoplasmic resistance, *R_cyt_*. Since there is no change in the resistance, the observed increase in capacitance that is occasioned by the drug mainly involves an increase in cell membrane capacitance. As capacitance (*C_mem_*) is directly proportional to the surface area (*A*) and inversely proportional to the distance (*d*) separating the charges, or the membrane thickness, we speculate that the increase in membrane capacitance is a consequence of drug (red circles) partitioning in a lipid membrane (Figure 4B). The resultant alterations in intra-bilayer interactions and the bilayer’s structural and functional composition lead to a decrease in bilayer thickness (*d*′).

According to Figure 2A, the initial change in membrane capacitance may be due to the attachment of amiodarone to the membrane surface, followed by drug partitioning and induced alteration in the membrane. Amiodarone, which has a strong lipid partitioning capacity (LogP = 7.9), also interacts with its known target protein, potassium channels, in the hydrophobic portion of the membrane. Initial increase in capacitance may be due to these initial events—molecular attachment, lipid participation, and interaction with potassium channels in intramembrane hydrophobic lipid–protein interface. As the drug is crossing the membrane and later distributed across the membrane during measurements, the capacitance may be returning to baseline or to another asymptotic limit as the membrane maintains equilibrium thickness. While we cannot infer, we can suggest that effects of lipid partitioning and interaction with proteins are initially localized in domains, since our measurements employ cells held in microwells exposing only a portion of the cell surface. The consequences of localized membrane-altering events on the cell surface are usually compensated through broadly distributed responses, as occasioned in membrane phase transition, bilayer bending, formation of lipid rafts, and curves. In fact, localized deformation of the lipid organization (which is also occasioned by drug partitioning at the protein boundary in the hydrophobic portion of the bilayer) brings about distributed changes to reduce the line tension, so cell membrane structure and function are not compromised, Formation of lipid and protein clusters indeed thickens the bilayer locally as compared to the surrounding membrane [28]. Since we are interested in the initial response of the cells to various drug doses, we did not continue the measurements beyond 1000 s.

We also tested acetylsalicylic acid/aspirin (10 µM) and glucose (5.5 mM) to see if these compounds affect cell membrane capacitance (Figure 2A). The molecules have distinct chemical, physical, and biological features. As with amiodarone, aspirin and glucose caused a sudden increase in membrane capacitance when they were introduced to the medium. However, the magnitude of the increase in capacitance was noticeably smaller for aspirin and glucose than the increase in capacitance occasioned by amiodarone. The aftereffects of amiodarone’s increase in membrane capacitance were also significantly different from aspirin and glucose. While aspirin- or glucose-induced increased capacitance returned to baseline (initial/pre-treatment) level in ~15 s, amiodarone-induced increased capacitance exhibited a gradual/slight decrease.

Furthermore, the gradual decline in increased capacitance followed a drug dose-dependent profile (Figure 2A) such that the decrease in capacitance was markedly slighter when the drug dose was higher. Extrapolation of graphs constructed from cell membrane capacitance measurements as a function of time indicates that curves may reach an asymptote at around 10,000 s (~2.5 h). One possible interpretation of these results is that amiodarone, which has a strong membrane partitioning, apparently stays in the lipid bilayer, leading to sustained/longer structural perturbation of the membrane. Glucose, acetylsalicylic acid, and amiodarone have different modalities of interaction with the lipid membrane. Their movement across the cell membrane is principally different; glucose moves mainly through facilitated diffusion, and aspirin moves readily through diffusion. Both movement modalities are thermodynamically driven across the plasma membrane from a higher concentration outside the cell towards the interior/cytoplasm of the cell. Amiodarone would also be subjected to thermodynamically driven diffusion forces; however, amiodarone can partition deeper into the lipid bilayer [29,30].

Moreover, a particular transmembrane protein segment is buried in the lipid membrane depending on the thickness of the bilayer as well as on the constraints imposed by the need to reduce the hydrophobic mismatch at the adjacent transmembrane segments and the local conformation of the side chains on the protein [31,32,33,34]. We reason that amiodarone partitioning in the cell membrane induces changes in membrane thickness, which can be attributed to hydrophobic mismatch due to partitioned drug in the bilayer’s core and resultant energy cost. Studies with amiodarone partitioning in lipid membranes substantiated that the presence of drug molecules in the bilayer causes bilayer deformation with increasing hydrophobic mismatch [35,36,37]. Furthermore, studies by Rusinova et al. indicate that the effect of amiodarone on potassium channel activity involves alterations in lipid bilayer thickness occasioned by drug partitioning in the bilayer [38].

## 5. Conclusions

We explored the utility of a microfluidic technique to collect real-time electrical impedance measurements on live cells that can be correlated with the biophysical properties of the cell membrane in response to interaction with drug compounds. The technique enables recording cell membrane capacitance and cytoplasmic resistance and provides data to study drug–membrane interactions. Our results show that membrane electrical capacitance is sensitive to changes occasioned by cellular interactions with drugs. We demonstrate that the portrayal of changes in cell membrane capacitance has the potential to assess the biophysical properties of the cell membrane in response to drug partitioning. The technique can provide a powerful approach to characterize specific and non-specific drug action in correlation with drug-induced alterations in the structural and functional organization of the cell membrane and membrane proteins. As various drugs can interact within the hydrophilic head region as well as the hydrophobic core of the cell membrane depending on their affinity, hydrophobicity, and partitioning, cell membrane capacitance measurements can provide supportive data for qualitative and quantitative assessment of drug–membrane interactions as well as the specific and non-specific effects of drugs on cells. The microfluidic system utilized in this study has the potential to measure changes in the electrical capacitance of the cell membrane on single cells. It promises a new approach to assess drug–membrane interactions and delineate specific and non-specific actions of the drug on cells.

## Figures and Tables

**Figure 1 micromachines-14-00316-f001:**
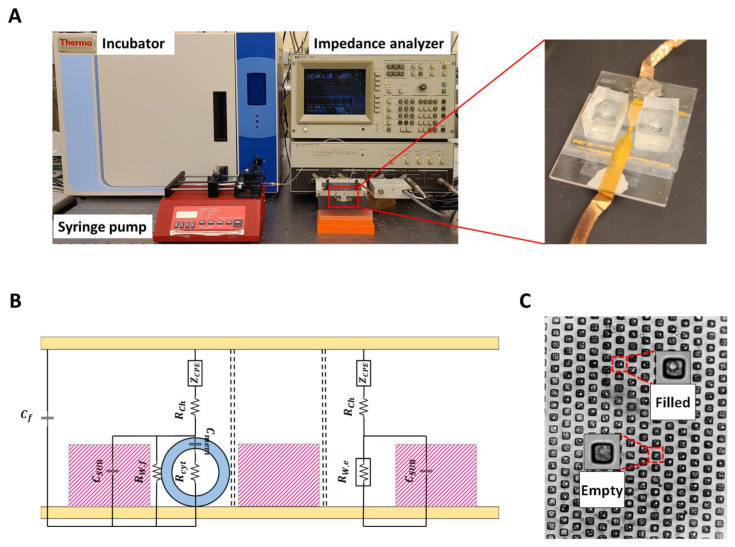
(**A**) Experimental setup and microchip. (**B**) Schematic of the circuit model that consists of all electrical components. (**C**) A view of the microchip showing the biological cells captured with gravitational force and empty wells.

**Figure 2 micromachines-14-00316-f002:**
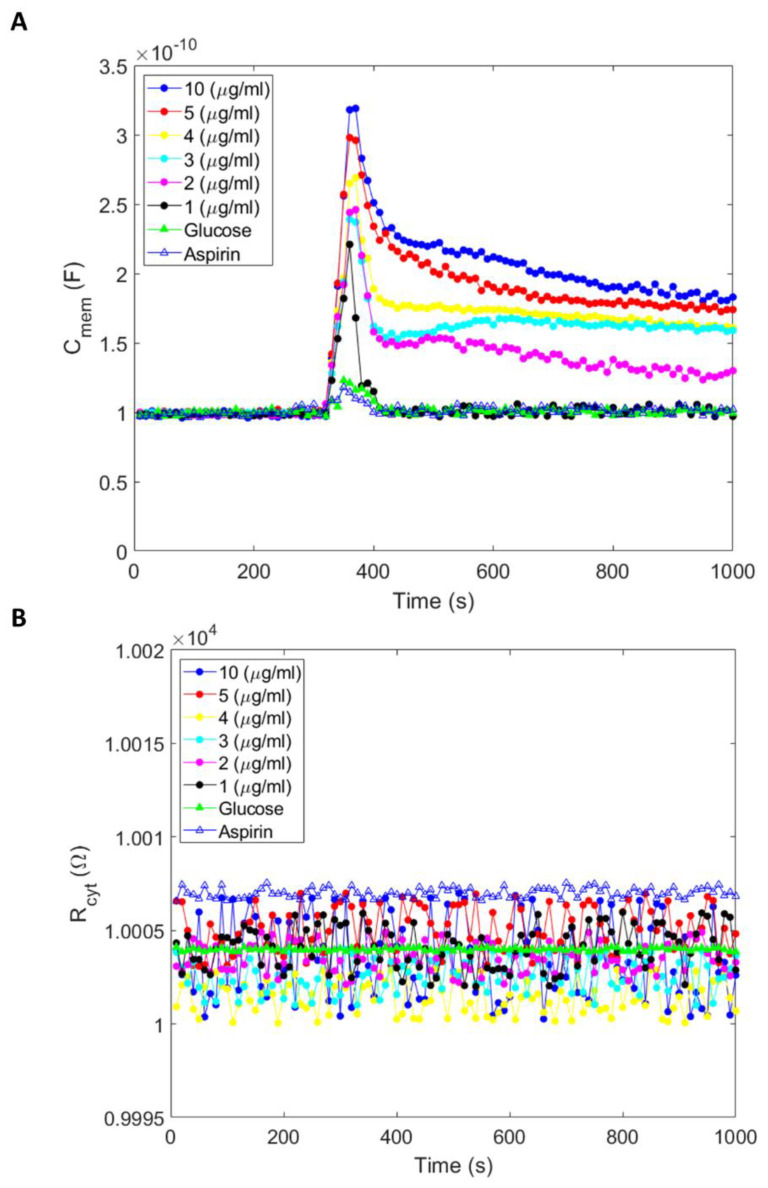
(**A**) Membrane capacitance and (**B**) cytoplasm resistance measurements were collected with PC-3 cells in response to amiodarone, aspirin, and glucose as a function of time. Measurements were collected every 10 s under glucose (10 µg/mL), aspirin (10 µg/mL), and various concentrations of amiodarone (1, 2, 3, 4, 5, and 10 µg/mL).

**Figure 3 micromachines-14-00316-f003:**
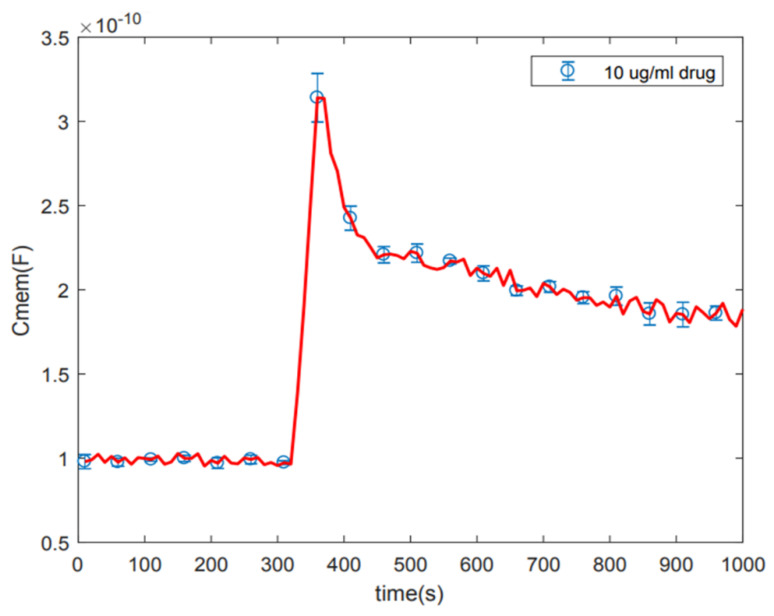
Standard deviation on 10 µg/mL amiodarone. The average value of three times repletion and standard deviation of the measurement for each five data points was plotted.

**Figure 4 micromachines-14-00316-f004:**
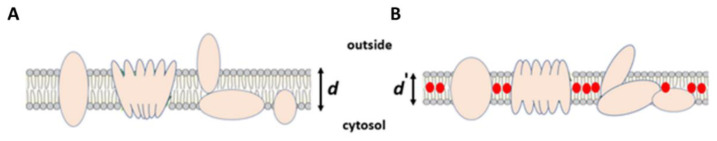
Plasma/cell membrane electrical capacitance is a biophysical parameter, which is related to the compositional and functional organization of the lipid bilayer. Drug partitioning in the lipid bilayer changes the properties of the cell membrane and alters membrane thickness. Changes in the membrane capacitance [(**A**) before applying the drug; red circles], corelates with drug partitioning [(**B**) drug–membrane interaction] in the lipid membrane and alters membrane thickness.

**Table 1 micromachines-14-00316-t001:** Sample of extracted growth media (GM) and growth with drug parameters (GMD) based on curve-fitting to the impedance spectra at different drug concentrations.

Solution	*R_ch_* (kΩ)	*R_w,e_* (kΩ)	*K*	*α*	*C_f_* (pF)
Growth media	33.430	16.121	2.28 × 10^−8^	0.698	6.21
1 µg/mL *amiodarone*	33.763	16.006	2.31 × 10^−8^	0.696	6.36
2 µg/mL *amiodarone*	33.656	16.081	2.35 × 10^−8^	0.694	6.28
3 µg/mL *amiodarone*	32.938	16.117	2.39 × 10^−8^	0.693	6.39
4 µg/mL *amiodarone*	32.641	16.381	2.43 × 10^−8^	0.691	6.48
5 µg/mL *amiodarone*	32.498	16.502	2.49 × 10^−8^	0.689	6.48
10 µg/mL *amiodarone*	32.056	16.978	2.76 × 10^−8^	0.681	6.52
5.5 mM glucose	36.021	16.893	2.81 × 10^−8^	0.729	6.13
10 µM aspirin	32.692	16.229	2.63 × 10^−8^	0.721	6.02

**Table 2 micromachines-14-00316-t002:** Extracted equivalent circuit and cell parameters before (CGM) and 70 s after (CGMD) introducing various drugs (corresponds to 400 s after starting the experiments).

Extracted Parameters	*R_ch_* (kΩ)	*α*	*K* (×10^−8^)	*C_f_* (pF)	*R_W,e_* (kΩ)	*R_W,f_* (MΩ)	*C_mem_* (pF)	*R_cyt_* (kΩ)
CGM	33.094	0.726	1.81	6.52	16.117	4.3282	98.91	10.006
1 µg/mL *amiodarone*	33.128	0.723	1.89	6.27	16.152	4.3196	115.08	10.003
2 µg/mL *amiodarone*	32.791	0.721	1.97	6.22	16.192	4.2889	158.19	10.005
3 µg/mL *amiodarone*	32.328	0.719	2.08	6.51	16.187	4.2112	162.21	10.002
4 µg/mL *amiodarone*	32.563	0.718	2.19	6.36	16.261	4.2357	189.09	10.005
5 µg/mL *amiodarone*	32.156	0.715	2.27	6.48	16.223	4.1922	234.12	10.001
10 µg/mL *amiodarone*	31.983	0.709	2.46	6.61	16.289	4.1261	251.01	10.004
5.5 mM glucose	35.398	0.746	2.18	6.32	16.324	4.3348	100.18	10.003
10 µM aspirin	31.083	0.738	2.39	6.17	16.112	4.3187	105.59	10.004

## Data Availability

Data in this work is available through the corresponding author.

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
