# Peer review of "Changes in Electrical Capacitance of Cell Membrane Reflect Drug Partitioning-Induced Alterations in Lipid Bilayer"

_micromachines, 2023, doi:10.3390/mi14020316_

Round 1

Reviewer 1 Report

In this manuscript, the authors tried to apply dielectric spectroscopy to noninvasively study electric properties of living cells in real time. The results are interesting, but their interpretation suffers from a very strong flaw.

In fact, the equivalent circuit model presented lacks membrane resistance, which is in parallel with membrane capacitance and may shunt the charges accumulated by capacitor.  In other words, changes in membrane impedance is influenced by both of them and observed evolution of impedance in the presence of amiodarone is nothing else than transient increase in membrane resistance, which leads to membrane depolarization – amiodarone is an inhibitor of potassium channels. It is evident that proper voltage across the membrane capacitor would influence its impedance. I would bet that perfusing cells with a buffer, in which sodium is replaced by potassium, would induce the same changes during dielectric spectroscopy experiments of the authors.

Moreover, the authors made no efforts to validate their results by some independent methods, which would undeniably show that their interpretation is flawed. The simplest example is that a living cell positioned in 30µm x 30µm x30µm well has a 30µm diameter at most, which corresponds to a maximal cell capacitance of about 30 pF in nonstimulated state (based on generally accepted specific lipid bilayer capacitance of 1 µF/cm^2) and it cannot be 100 pF as the authors claim. The value of 30 pF is the most common membrane capacitance reported for PC3 cells in the literature (see for example Ashmore et al, 2019, J. Physiol).

Furthermore, no literature to my best knowledge suggests such drastic changes in membrane thickness and, consequently, capacitance at the level of the whole cell in the presence of a drug, similar to what the authors try to present.

All above suggests that the authors should profoundly re-consider their model, which with the addition of membrane resistance may become non-trivial to solve. However I encourage the authors to do so, because dielectric spectroscipy may potentially become useful in noninvasive screening of ion channel inhibitors.

Reviewer 2 Report

This work studied the interaction of three different drugs with cell by measuring and extracting the changes of electrical capacitance of cell membrane using a home-made electrochemically microchip. Drug-induced membrane capacitance changes were successfully extracted, and assigned to the lipid partitioning properties., which may be reflecting the different interaction mechanism of different drugs. I would like to see this work published, but a minor revision is necessary.

1.     Too much self-citation in the Introduction part.

2.     I would like to see the repeatability of different microchips that fabricated in different batch.

3.     Please explain why the capacitance increased by addition of drugs all returned to baseline. The drug concentration was maintained during the whole measurement.

4.     It’s better to include another small molecule which is widely accepted that could penetrate into lipid bilayer.

5.     Several ion channels (especially, potassium channel) could be inhibited by Amiodarone. Why no apparent change in overall cytosolic resistance was observed?

6.     If the drug molecule only binds on the membrane (with membrane proteins, or polysaccharide or lipid) without penetrate into lipid bilayer, how the membrane capacitance would change?

7.     Glucose goes through the cell membrane by active transport, not thermodynamical diffusion.

8.     The cell activity after the measurement should be confirmed.

Reviewer 3 Report

In the present study, microfluidic impedance spectroscopy technique developed bu Prof. Beskok’s group has been applied to discuss responses of PC-3 prostate cancer cells to different doses of amiodarone, aspirin and glucose. The experimental procedure is explained, and real time data for the capacitance and resistance of the cell membrane is presented. The discussions are supported by the results. I believe this study is quite beneficial for the scientific community. It is publishable after minor revision. Here are my detailed comments:

1. The parameters are extracted from the experimental data, using in-house MATLAB code. It would be beneficial for the readers at least to discuss what kind of parameter estimation algorithms are utilized in the code. What would be quality of the fit (is there any R2 parameter and/or some uncertainty associated with the estimation)?

2. It may be better to give the Table 2 in the form of Table 1, or Table 1 in the form of Table meaning that columns are the dose of different drugs or the rows are  the dose of different drugs.

3. For a better explanation of the experimental setup, the name of the instruments can be included in Figure 1.

4. The resolution of Figure 2 can be improved, and the x-axis may start at zero and end at 1000. It may be beer to use the same format in Figure 3 (same axis name, same ticks, same units, same color for the corresponding curve). Only repeated experiments are available for 10 ug/ml? May be combining Figure 2 and 3 may be an option. The uncertainty values in Figure 3 may be also tabulated at the indicated data points.

Round 2

Reviewer 1 Report

I regret to see that the authors did not take seriously my comments. My criticism remains the same:

11.  Stationary membrane voltage across capacitor in the model is the major actor determining the membrane impedance. Membrane resistance (lacking in the model and though too large to influence the impedance directly) may shunt this membrane voltage during experiment - the principal action of amiodarone is to inhibit potassium channels and thus to depolarize the membrane – membrane voltage approaches zero, which obviously would change impedance of membrane capacitance.

22. A living cell positioned in a 30x30x30 µm cube cannot have capacitance more than 30 pF. The authors cannot cite papers, in which only LNCaP cells were used [3,5,6], these are bigger cells than PC3. In the reference [4], the median value for PC3 cells is 40 pF, while 100 pF is rather an extreme value for gigantic cells that would not otherwise fit to the above cube. And reference [7] is about myocytes and not PC3 cells.

33.  The explications of how the membrane capacitance may increase by 50% are not convincing. I do not see how partitioning of amiodarone would decrease membrane thickness by 33%, this is enormous change. Moreover, the binding of amiodarone on the surface of the membrane may indeed increase apparent surface on the external side of the membrane, but it does not mean that bilayer surface changes accordingly. And again, no available literature supports that such drastic changes in membrane capacitance are possible.

Reviewer 2 Report

I concerns have been properly  addressed, and I recommend publication of this manuscripts.

Author Response

Thank you for your constructive suggestions, which significantly improved the quality of our manuscript.